# Understanding the Benefits, Challenges, and the Role of Pet Ownership in the Daily Lives of Community-Dwelling Older Adults: A Case Study

**DOI:** 10.3390/ani11092628

**Published:** 2021-09-07

**Authors:** Nataša Obradović, Émilie Lagueux, Karine Latulippe, Véronique Provencher

**Affiliations:** 1School of Social Work, Faculty of Arts, Humanities and Social Sciences, Université de Sherbrooke, Sherbrooke, QC J1K 2R1, Canada; 2Research Centre on Aging, Centre Intégré Universitaire de Santé et de Services Sociaux de l’Estrie—Centre Hospitalier Universitaire de Sherbrooke (CIUSSS de l’Estrie—CHUS), Sherbrooke, QC J1H 4C4, Canada; 3School of Rehabilitation, Faculty of Medicine and Health Sciences, Université de Sherbrooke, Sherbrooke, QC J1K 2R1, Canada; emilie.lagueux@usherbrooke.ca; 4Research Centre, Centre Hospitalier Universitaire de Sherbrooke (CRCHUS), Sherbrooke, QC J1H 5N4, Canada; 5School of Physical & Occupational Therapy, Faculty of Medicine and Health Sciences, McGill University, Montréal, QC H3A 0G4, Canada; karine.latulippe@mail.mcgill.ca

**Keywords:** community-dwelling older adults, companion animals (CA), pet ownership, healthcare provider, activities of daily living

## Abstract

**Simple Summary:**

This qualitative case study explores the perceived benefits and challenges of pet ownership for a community-dwelling older adult and her miniature schnauzer, from the perspectives of the pet owner and her community healthcare provider. The findings suggest that the pet’s well-being is an important part of the pet–owner relationship for Violet, the older adult. Sharing her daily life with her pet, Jack, gives her a sense of safety, positively influences her mood, and motivates her to carry out her daily activities. In return, Jack benefits from her daily presence and care. The challenges of pet ownership included a low-risk potential of falling, pet-related concerns, and financial costs. Both participants agree that the benefits outweigh the challenges for both Violet and Jack. Findings of this study suggest that caring for the pet is a meaningful aspect of the pet–owner relationship. Future studies should explore how to support human–animal relationships for community-dwelling older adults via pet ownership. Improving the fit between characteristics of the elderly pet owners and their pets will maximize benefits and minimize potential challenges; thus, supporting both aging-in-place and the well-being of animals.

**Abstract:**

Human–animal interactions may positively impact the health and well-being of older adults. Considering about one third of community-dwelling older adults report owning a pet, better understanding the benefits, challenges, and the role of pet ownership may help support the relationships between older adults and their pets. This case study aims to better understand the role of pet ownership in the daily lives of older adults and explore the benefits and the challenges of owning a pet for this population. Interviews were conducted with Violet, a 77-year-old dog owner and her healthcare provider. Qualitative data were analyzed by two evaluators and validated by the participants. Both participants agree that the benefits outweigh the challenges for both the older adult and her pet. The benefits and challenges were the following: Violet, taking care of her dog (Jack), (1) could provide Violet with a sense of safety and positively influence her mood; (2) may introduce a slight fall risk; (3) includes financial costs to consider. Ensuring Jack’s well-being is important for Violet and her dog benefits from Violet’s continual presence and care. The findings suggest that improving the fit between characteristics of the owner and their pet may support the meaningful role of pet ownership within the context of aging-in-place.

## 1. Introduction

Older adults aged over 65 years account for 17.5% of the Canadian population; this percentage is expected to exceed 23% by 2036 [1,2]. Almost 20% of older adults report feelings of social isolation, which is known to adversely affect their psychological well-being and their physical health [3,4,5,6,7]. Human–animal interactions (HAIs) may be an avenue worth exploring to support the health and well-being of this population, as research about the positive psychological, social, and physical impacts of HAIs is promising [8,9,10,11,12,13,14]. Considering about one-third of community-dwelling older adults report living with at least one companion animal (CA) [15], pet ownership might play an important role in supporting aging-in-place.

Relating to psychological and social benefits, recent studies suggest that pet ownership may improve well-being, life satisfaction, and happiness, as well as decrease loneliness and social isolation, depressive symptoms, and anxiety [16,17,18,19,20,21,22,23,24]. It may also increase levels of physical activity and/or walking of older adult pet owners [25,26,27,28,29]. Other benefits include providing an overall sense of purpose and encouraging a daily routine [24,30,31]. Apart from these benefits for older adults, animal well-being is also important to consider in the pet–owner relationship. Providing daily care to a CA to ensure its well-being entails ongoing duties and responsibilities, regardless of the pet owner’s age. Pitteri and colleagues report that dogs owned by older adults have similar physical conditions to dogs owned by adult owners. However, the study suggests that the dogs’ quality of life may be influenced by contextual factors, such as older adults’ employment conditions, level of education, and type of dwelling [32].

Other potential factors to consider are health conditions and functional decline that may occur as part of normal aging and that affect older adults more frequently. These may exacerbate some challenges associated with pet care [9,33]. Furthermore, the companion animal’s health conditions and needs may change (e.g., needing physical assistance to use the stairs, more regular veterinary visits), which may modify their level of needed care as they, too, age [32]. Frequently reported challenges related to owning a pet by older adults include: grief related to pet loss and fear of outliving them, pet care being perceived as a chore [9,10,13,17,23,34], risk of falls [35,36], financial costs (especially fees related to veterinary care), and the fear of needing to leave a pet in the event of a relocation [34]. If such challenges become too great for the older adult pet owner, the well-being of the owner and the companion animal may be compromised if the owner struggles to fulfill the basic needs of both parties.

Therefore, considering these potential issues, it is essential to depict a realistic representation of the benefits, challenges, and the role of pet ownership in the daily lives of community-dwelling older adults. Optimizing benefits while minimizing challenges will ultimately support the health and well-being of both older adults and their CAs. However, there is little research on if (and how) the potential benefits outweigh the challenges of owning a pet for community-dwelling older adults and their CAs [11,14].

As most older adults prefer to age in their homes [37], community-based healthcare services may play a crucial role in enabling older adults and their CA to age-in-place. Indeed, besides promoting independent living and safety in their homes, healthcare providers may also support pet–owner relationships. Yet, few studies have simultaneously explored the perspectives of both healthcare providers and older care recipients on the benefits, challenges, and the role of pet ownership in their daily lives [14]. Combining these perspectives could provide a comprehensive and detailed view of how pet–owner relationships may simultaneously promote healthy aging-in-place of community-dwelling older adult pet owners, as well as the well-being of their CAs. For this study, community-dwelling older adults included older adults living in their homes or in an assisted living facility, but excluded those living in nursing homes (i.e., needing continuous medical care).

Thus, the purpose of this study was to further explore the role of pet ownership in the daily lives of community-dwelling older adults. More specifically, it aimed to: (1) describe the benefits and challenges of owning a CA for both older adults and their companion animals, as perceived by older adults and their healthcare providers; (2) explore the role of pet ownership in the daily lives of older adults; (3) examine the balance between the benefits and the challenges of owning a pet by this population.

## 2. Methods

### 2.1. Study Design

A qualitative, single-case study was conducted. The case involved (1) a community-dwelling older adult and her companion animal (CA); (2) her community healthcare provider for home care services. Case studies allow for in-depth, multi-faceted explorations of complex and emerging phenomena and issues [38]. In the current study, aspects of pet ownership were explored in relation to the characteristics of the person, their CAs, and the environments in which they evolve [39].

### 2.2. Recruitment

Participants were recruited by purposive sampling through a professional organization. The healthcare provider who participated in the study worked in a community health centre; she referred to the first author, (N.O.), an older adult from her caseload who she judged could meet the objectives of this study. This case enabled the examination of how an older adult pet owner managed the demands of pet ownership despite potential challenges, such as physical disabilities, and despite being the only caretaker of her pet. The study was approved by the CIUSSS de l’Estrie-CHUS Ethics Committee (#2020-3336) and informed consent was obtained from both participants.

### 2.3. Participants

The older adult participant is a 77-year-old woman named Violet (pseudonym) who lives with her dog, an 8-year-old miniature schnauzer named Jack (pseudonym). Jack is Violet’s companion animal and has not received any training to acquire specific skills (i.e., he is not considered a service dog or emotional support animal). Violet does not receive any financial support or services related to her disabilities to help with Jack’s care. As for Violet, she has physical disabilities and moves around in a motorized wheelchair, which is the main reason that she receives community-based healthcare services. Violet’s community healthcare provider has more than 15 years of experience in the field and, at the time of the study, had known Violet for the past 17 years. She is also Violet’s healthcare coordinator, meaning that she plans and organizes her medical and health services. She reassesses Violet’s needs annually to ensure that the home care services are adapted to suit her needs.

### 2.4. Data Collection

The first author conducted semi-structured interviews with Violet and her community healthcare provider. Both interviews aimed to explore the perspectives of Violet and her healthcare provider about themes related to pet ownership (psychological, physical, daily benefits, and challenges of taking care of the CA, the role of pet ownership in daily life, the pet–owner relationship, and the well-being of the pet). Interview guidelines were developed by the research team and feedback was obtained by an experienced healthcare provider working with older adults in a clinical setting. Table 1 provides examples of some interview and follow-up questions for both participants. Prior to the interviews, both guidelines were tested with individuals who met the inclusion criteria, i.e., two community-dwelling older adult pet owners and a community healthcare provider. Two interviews were carried out in April 2020 via videoconference with the healthcare provider and by telephone with the older adult, due to the COVID-19 (coronavirus disease 2019) pandemic context. Each interview lasted from 60 to 90 min. As it was not possible to conduct the interviews at the older adult’s home, as initially planned, the participants were questioned about the home environments and interactions with the CA, including its behaviour (i.e., does it jump up on the older adult, does it bark, etc.). Sociodemographic data about age, gender, education, years of clinical experience for the community healthcare provider and pet characteristics (age, species, number of years living with the older adult, pet care activities, health condition) were also collected during the interviews to provide contextual information. This information is reported in Section 3 to the extent that the confidentiality of the participant was kept.

### 2.5. Data Analysis

Interviews were audio-recorded, and the transcripts were read numerous times to get a sense of the data. Each transcript was analyzed line-by-line independently using thematic content analysis (continuous thematization) by the first and third authors, who have extensive experience in qualitative analysis [40]. Units of analysis were the older adult and the healthcare provider’s perceptions about benefits, challenges, and the role of pet ownership in daily life. Themes emerged during the reading of the transcripts. The authors met to review and to compare themes for similarities and/or discrepancies. In the case of disagreements, a consensus was sought. The themes and their relationships were then categorized within a matrix containing categories pertaining to the person/companion animal, the environments, and the activities of daily living [39]. This matrix was used to determine the overall balance between the perceived benefits and challenges related to pet ownership. The content of written summaries was validated by conducting two additional interviews with the participants, lasting from 60 to 90 min. This step (member checking) increases the credibility and reliability of the analysis and interpretations [41]. In addition to the two perspectives of the participants, the independent identification of themes by two evaluators ensured data triangulation, and memos written by the first author further increased the credibility of the analysis [41,42]. Furthermore, the research team regularly discussed the findings to maintain reflexivity and to gain a deeper understanding of the participant’s characteristics, daily functioning, and context.

## 3. Results

### 3.1. Case Summary

Violet and her dog Jack live in an urban area, in a one-bedroom rent-subsidized apartment. Her apartment is subsidized due to having limited financial resources. Jack was offered to Violet as a gift from her daughter, a couple of months after Violet’s former dog passed away. Violet is Jack’s sole caretaker, and she has raised him since he was a puppy. Jack is a small adult dog, weighing approximately 12 lbs, and he is sturdy, according to Violet. She reports that these characteristics are an adequate fit for her and enable her to maintain her balance while picking him up. Furthermore, Violet mentions that Jack is in good health and that she ensures that she meets his basic daily needs. Violet, divorced for several years, maintains regular contact with her children and grandchildren. Before the pandemic, she visited them and engaged in activities such as shopping with her daughter. Violet has loved animals since childhood and has had dogs as pets for all of her adult life.

Violet has physical disabilities, which are mainly due to a medical condition that resulted in having multiple amputations to her lower and upper limbs over the years. She wears below-the-knee leg prostheses and reports having daily lower back pain, which varies in intensity. According to the PRISMA-7, a screening tool that identifies older adults at risk for moderate to severe disabilities, she has significant disabilities [43]. According to her healthcare provider, Violet has adequate cognitive functioning, which was also the case during the interviews (i.e., she showed no signs of cognitive impairment according to the clinical judgement of the interviewer).

Regarding Violet’s functioning during daily activities, she carries out some activities independently and needs assistance for others. She has been receiving community-based home care services for several years, including help for self-care, housekeeping, and maintenance for her motorized wheelchair. However, she can prepare meals, manage her budget and appointments independently, move around inside and outside of the home, as well as carry out leisure activities. Inside her apartment, she walks with leg prostheses. She leans on surfaces, such as furniture or walls, to maintain her balance. According to her healthcare provider, her environment is adapted to her needs, which helps her to live independently and safely in her home with her dog. In the community, Violet always uses her motorized wheelchair for shopping with Jack or going to medical appointments. She reports feeling safer this way. For leisure, she enjoys playing games on her computer and spending time with Jack. Lastly, she takes care of her dog independently. Caring for Jack involves feeding him, brushing him, and taking him on daily walks. Violet reports that she has bathed him occasionally in the past, but she now prefers to take him to the groomer because she feels that it is safer than manipulating him in the bathtub, due to her back pain. Violet and Jack’s daily walks are adapted to the seasons (i.e., they stop during the wintertime and resume in the spring). As for Jack’s healthcare, Violet takes him to the veterinarian only when she deems it necessary. She reports that Jack has been healthy since he was a puppy, except for occasional ear infections. According to both Violet and her healthcare provider, people who meet Jack seem to appreciate him and they did not raise any issues regarding his behaviour, other than him occasionally being persistent when demanding affection.

### 3.2. Benefits and Challenges of Pet Ownership for the Older Adult

Several benefits and challenges were mentioned by both Violet and her healthcare provider. They are presented here according to psychological and physical benefits/challenges.

#### 3.2.1. Psychological Benefits and Challenges

*Sense of security and good mood.* The main finding regarding the psychological benefits is that Jack’s presence in Violet’s daily life has a positive influence on her emotions—an aspect that both Violet and her healthcare provider agree on. Violet mentions that Jack makes her “feel safe” and helps her “be in a good mood”. She reports that without him she would feel “alone and abandoned”. Having a presence in her home and this sense of security is very important for Violet. This presence in her house became even more important to her since she has been living alone and after she turned 60:

“I feel safe, in the house. That’s very important for me because I don’t like the darkness, I don’t like when there is wind, when it thunders (laughter). So, Jack makes me feel safe, you know? […] At least I’m not alone in the house, there is a presence.”

According to both Violet and her healthcare provider, Jack also helps Violet channel stress and anxiety, which helps calm her down and weather difficult emotions. They both agree that caring for Jack is a direct factor that contributes to regulating Violet’s mood, since tending to her dog’s needs focuses her attention and takes her mind off negative emotions. Jack also provides feedback with his behaviour, which allows Violet to be more mindful of her emotional state and to regulate her emotions, for both her own and Jack’s well-being:

“When I’m not calm, when I’m under continuous stress, Jack feels it and he becomes bothersome. […] When I’m like that, he senses it and wants to cuddle, cuddle, cuddle. So when I see that I’m making him unhappy, I say to myself: Calm down, look at what you’re doing to him. He was happy before and now he’s feeling down because of you. So that’s why I have been telling you: Jack is a big part of my life!”

*Source of pride.* Besides positively influencing Violet’s moods, Jack is also a source of pride. Violet considers that taking care of her dog is akin to taking care of a child, and that, like a child, he needs “to go outside and get fresh air, […] needs someone to take care of him, to give him everything”. She therefore ensures Jack’s well-being and prides herself with her ability to appropriately care for him, despite her disabilities:

“I feel proud of myself! A lot of people ask me: How do you manage to keep a dog in your house—you have no hands or legs. And I answer: So what? Do you need hands and legs to take care of a dog? No! You manage and find ways to do it. I don’t have any problems with my dog. I brush him, I bathe him—you find ways to get organized. There are a lot of things that you figure out along the way.”

In a sense, providing care for Jack and being able to meet his needs gives Violet the opportunity to be a care provider, a role that she feels comfortable in and that she has carried out her entire life. This aspect was only mentioned by Violet.

*Concerns about health and the future.* Both Violet and her healthcare provider agree that concerns related to Jack’s health or his future are some of the main challenges. They both considered what would happen to Jack in the event of a hospitalization or a relocation, where Violet might have to part with her dog:

“If he’s healthy like he is now, and I leave for the hospital and they send me to a nursing home, I can’t bring him with me. What do I do then? My son said that he was going to take him in, and I agree that he won’t be neglected by my son! But I won’t have him beside me like I do right now…To get rid of Jack and give him to my son, my life would be over…that, would bother me.”

Both participants agree that being obliged to part with Jack under such circumstances would be difficult and would bring up negative emotions. In Violet’s words:

“I would tell myself: You’re abandoning him. […] Say I don’t have a choice and I can’t take care of him anymore…I’m quite scared to go there [nursing home] and not be able to bring him with me because there aren’t any [animals]. At my age, that bothers me a little more, [the] I think about it…”

In anticipation to departing with Jack, Violet even mentioned that he might be the last dog that she shares her life with, to avoid such difficult feelings.

#### 3.2.2. Physical Benefits and Challenges

It is worth mentioning that the only benefit regarding pet ownership and Violet’s physical condition was reported by the healthcare provider. According to her, he may encourage Violet to be slightly more active in her home (e.g., bending down to feed him, walking around in the home to find him). However, as most of the walks are done with the motorized wheelchair, she reports that, in Violet’s case, benefits usually associated with walking are likely not significant. Nonetheless, Jack encourages her to go out into the community and get fresh air.

*Risk of falls.* A potential risk raised by Violet’s healthcare provider is the risk of falling. Although she assessed the risk as low, she acknowledged that it is still present. For example, in situations when Violet walks with her prostheses or bends down to pick up Jack:

“It could put her at risk of falling when she moves quickly like that […]. When she picks him up slowly to feed him, there is no danger, but maybe with the excitement, when she picks him up quickly and says: ok, you’re being a pest…she could fall.”

Violet reports that Jack has never been the cause of a fall and perceives the risk of falling because of him as being null. She points out that Jack has adapted his behaviour to her health condition. For example, he jumps up on her motorized wheelchair instead of her picking him up like in the past. Violet also mentioned that the dog’s size is well suited to her lifestyle and habits, that he walks in front of her, and stays out of her way when she walks around in her apartment. Lastly, Violet takes her dog outside only with her motorized wheelchair, weather permitting, and does not pick him up when she feels tired. Moreover, Jack uses puppy pads inside the home, which enables Violet to take care of her dog’s needs independently even when she cannot take him outside. She does admit, however, that you cannot predict the future and that it is not possible to assert that a fall will never occur.

#### 3.2.3. Other Potential Challenges

*Financial costs.* Both Violet and her healthcare provider mentioned financial costs surrounding pet care as a potential issue, especially regarding veterinary fees. However, their perspectives differed slightly when reporting them. Her healthcare provider points out that veterinary costs could be “a financial burden for her […] maybe she sacrifices some things to be able to pay for the veterinarian services.”

For Violet, financial costs do not seem to be a current issue and she reports not having to sacrifice her own well-being to meet Jack’s needs. She explains that she takes him to the veterinarian only when he is sick and tries to find affordable care solutions, if necessary. For example, after many costly visits to the veterinarian to treat Jack for an ear infection, she followed recommendations from a previous veterinarian and instead bought less expensive eardrops for children, which helped treat him successfully. Violet’s cost-reducing solution does not necessarily confirm her healthcare provider’s assumption that “there is probably a part of her budget that is for the veterinarian”. However, it is an example of how she manages expenses related to pet care and how this enables her to take care of Jack despite her limited income.

*Relationships with healthcare providers.* Violet’s healthcare provider recalled a situation where there were tensions between Violet and a community home health aide. The home health aide was newly appointed to Violet and was not comfortable with Jack’s presence, even though he was enclosed in his cage, in another room. Both Violet and her healthcare provider agreed that Violet followed the home support organization’s regulations by confining her dog in another room. They also agree that it is necessary to follow these regulations if a healthcare provider is uncomfortable with Jack’s presence. However, her healthcare provider reports that usually the home health aides appointed to Violet appreciate Jack. Lastly, Violet mentions that she has never refused home support services and that she would not refuse them for this reason.

### 3.3. Well-Being of the Companion Animal

From both Violet and her healthcare provider’s perspectives, Jack received adequate care from Violet. One of the major advantages reported by Violet relating to her dog’s well-being is that Jack always has someone with him, since Violet spends most of her time at home. The healthcare provider suggested that perhaps Jack could benefit from going on more walks during the wintertime, which could, according to her, further improve his well-being. Nonetheless, both participants report that Jack’s basic needs were fulfilled according to them, and that his overall health and well-being were assured under Violet’s care.

### 3.4. The Role of the Pet–Owner Relationship and Owning a Pet in Daily Life

#### 3.4.1. Meaningful Activity

Both Violet and her healthcare provider agreed that the pet–owner relationship between Violet and Jack was significant; taking care of her dog is a highly meaningful activity for Violet, even central in her life. Violet considers Jack “like her baby” especially since her children have moved out of the home and that she lives alone. Her healthcare provider notes that Violet is not socially isolated and that, on the contrary, she maintains good relationships with her family members. In her opinion, taking care of Jack is not only an activity that Violet engages in due to social isolation or feeling lonely; he is a daily companion and, in Violet’s words:

“I have something to take care of. Me, I need something to keep busy, and well Jack, he’s that. I can watch television and then I say: Ok, we will go take a nap on the couch then watch television. I always need to talk to him. And then Jack jumps up on the couch […]. He sleeps, and I watch television, but he is right beside me. You know what I mean? I’m not alone in the house, there is another presence.”

#### 3.4.2. Providing Purpose and Routine

In addition to these perceived psychological benefits, according to her healthcare provider, caring for Jack provides Violet with a purpose and a daily routine, and gives her a daily structure:

“Well the fact that she can’t decide to not get up one morning or take him out, it forces her to get up every morning—it gives her structure.”

When describing a typical day in Violet and Jack’s lives, as well as the daily activities that Violet carries out, clearly Violet and Jack’s daily routines are integrated. In Violet’s words, life without Jack would be, “boring, very boring! That wouldn’t be any kind of life.” She adds:

“What kind of life is that for an older person, always alone? I realize that even more since I turned 60.”

For Violet, taking care of her dog is an impetus for carrying out daily activities, such as eating, shopping, and going out into the community to take walks:

“Every day, we get out, and he walks 8 km per day. And I take him out on walks every day—otherwise I wouldn’t go outside, I would stay at home.”

The role of pet ownership in Violet’s daily life was clear when she spent five months living without a pet after her late dog passed:

“When I lost my other little dog to cancer, I didn’t do anything for five months. I didn’t feel like cooking—I loved cooking—I didn’t feel like it anymore. I didn’t feel like…oh the real word is living. I don’t know if I would still be here today, because after three months Jack entered my life and I started taking care of him. That’s when my smile and mood came back, everything came back!”

Her relationship with Jack significantly enhances her engagement in other daily activities, such as going out into the community, shopping, interacting with other people, and taking care of herself. Indeed, Violet and Jack go shopping together, and Jack facilitates social interactions as he can stay in her motorized wheelchair. Violet reports that engaging in these activities shows her “her independence.” When a life without Jack was mentioned during the interview, Violet promptly answered that she would not carry out some of her activities if it were not for his presence:

“I wouldn’t go, I wouldn’t be interested in going anywhere […] not even going to see my children. You know, Jack is kind of like my husband. You don’t go out without your husband—well I don’t go out without Jack. I wouldn’t go grocery shopping, I would ask my children to do that stuff, like they do now [during the COVID-19 pandemic].”

Besides going out for walks with Jack and into the community, Violet also describes how her relationship with her dog encourages her to take care of herself so that she is able to be there for him as long possible. For example, having to feed Jack everyday prompts her to have dinner as well, as they frequently eat together. Meeting her dog’s needs and his well-being are clearly important for Violet. Although she is sometimes concerned about Jack’s future if something happened to her, Violet reports that she “will focus on Jack” and that:

“I don’t want to get sick. I will be careful and protect myself in any possible way. I have about four or five years left with him, so I tell myself: Don’t fool around, you can tough it out five years!”

Her healthcare provider also illustrates the positive role of pet ownership and the pet–owner relationship in Violet’s life:

“For Violet I think it’s positive—because there could be a negative side to being obligated to follow a certain routine for the dog. If someone wanted to do other activities or wanted more flexibility…but for her it’s positive. She has the time to do it, and I don’t think it prevents her from doing things that she would do if she didn’t have a dog.”

### 3.5. Balance between the Benefits and Challenges of Pet Ownership

Both Violet and her healthcare provider agreed that the benefits of pet ownership outweigh the potential challenges for both the older adult owner and her companion animal. On one hand, in Violet’s case, the pet–owner relationship with Jack has overall psychological benefits, adds meaning to her daily life and enhances engagement in daily activities. On the other hand, Jack benefits from Violet’s daily presence and care. Both participants perceive that the potential challenges associated with pet care are manageable by Violet and that she can adequately fulfill Jack’s needs to ensure his well-being (e.g., feeding him, taking him on walks). They both agree that pet ownership is a positive experience in this case, despite factors that could have potentially increased the demands associated with pet care (e.g., Violet being Jack’s sole caretaker, having physical disabilities, limited financial resources). In other words, from their points of view, the pet–owner relationship is beneficial for both the older adult and her companion animal’s well-being.

## 4. Discussion

The aim of this study was to explore the role of pet ownership in the daily lives of community-dwelling older adults from the perspectives of an older adult and her community healthcare provider. To our knowledge, this is the first study that aimed to gain a comprehensive view of the interaction between the characteristics of a person and her CA, their environments, and their daily habits, to explore if (and how) the benefits and challenges of pet ownership outweigh one another.

One of the main findings in the current study is that the pet–owner relationship is highly meaningful for Violet, the older adult participant, and that pet ownership plays a central role in her daily life. Both the older adult and her healthcare provider concluded that, in Violet and Jack’s case, the benefits of the pet–owner relationship outweigh the potential challenges for both parties, despite the owner having physical and functional limitations. On one hand, they highlighted that Jack’s continual presence is beneficial for Violet’s psychological and physical health and that the responsibilities associated with fulfilling her dog’s needs keep her busy. On the other hand, Violet’s presence, and ability to provide adequate care to Jack is also beneficial for the companion animal. She takes pride in being able to meet her dog’s needs, which gives her a sense of independence. Jack is an integral part of his older adult owner’s daily life, which is in line with reported findings [23,44]. His companionship makes Violet feel safe and less alone—these aspects have gained even more importance for her over time. Daily companionship is indeed one of the main reasons reported by older adults for adopting pets [10,13,45,46]. CAs are often considered like family members [31,46], which is in line with the findings of the current study. Violet often referred to Jack as her “baby” or “like a child” that needs to be cared for. Both Violet and her healthcare provider acknowledge the importance of Jack’s daily companionship. Scheibeck and colleagues (2011) also acknowledged that dogs can be an important part of community-dwelling older adults’ lives, partners in life, and companions providing their owners with a sense of purpose and daily structure [31]. It is interesting to note that Violet’s healthcare provider does not consider Violet as being socially isolated and does not perceive pet ownership as a way to compensate a lack of social relationships, in Violet’s case. Such findings may indicate the importance of the human–animal bond that is formed through pet ownership, despite an older adult being socially connected. Human–animal bonds may fulfill different needs in daily life, even when an older adult maintains social relationships, which is an aspect that merits further research. Jack and Violet’s relationship has indeed some characteristics of the human–animal bond, in that it is continuous, reciprocal, and both parties mutually benefit from an increase in their well-being [47].

Regarding the challenges of pet ownership, a risk of falling while walking in her home was the main concern, considering Violet’s leg amputations. However, both participants assessed this risk as being low, and no falls related to Violet’s dog were reported. Violet and Jack’s case may be an example of how an adequate fit between the characteristics of the older adult, their CA and their environment may maximize the benefits related to pet care. Violet has adapted her routine or found solutions to minimize the challenges and to be able to meet Jack’s needs, despite her physical disabilities and limited income. Jack also seems to have adapted his behaviour to Violet’s health condition. According to Violet, her dog’s physical characteristics (i.e., small size, sturdy stature) fit well with her own characteristics. This may also play a role in managing the fall risk. In this case, the risk of falling is considered low, but as fall risk assessment is complex and multifactorial, it should be noted that it might not be the case for other older adults that share their lives with companion animals. Indeed, older adult pet owners may present different protective and risk factors (e.g., functional limitations, pain, depressive symptoms) [48], which may influence the fall risk. In a study conducted with 16 cases of pet-related falls involving older adults aged over 75 years, Kurrle and colleagues (2004) reported that some falls were related to the person’s behavior (e.g., climbing on a chair to catch a pet canary), while others were related to the animal’s behaviour (e.g., dog pulling on a leash or tripping over a cat in a dark hallway) [35].

However, to our knowledge, analyzing the fit between the requirements of pet care, the older adult pet owners, their companion animals, and their environments, has seldom been explored. Future research may further investigate contextual factors when assessing the fall risk of older adult pet owners. Such research may help to identify protective and/or risk factors, which managed, may decrease the risk of falling. Assessing such factors may be a way for healthcare providers to support human–animal relationships through pet ownership [48]. This could help determine if, how, for whom, and under which circumstances pet ownership may be beneficial for older adults with disabilities and their companion animals, in the context of aging-in-place. In cases where owning a pet appears to be harmful for either the well-being of the older adult or the pet, interventions could be implemented to reduce the challenges associated with pet care and enhance the well-being of both parties (e.g., helping find animal-friendly housing, offering assistance with pet care) [49,50,51]. Such interventions may be especially beneficial for older adults with disabilities, socioeconomically vulnerable, frail, or isolated older adults, for whom it may be more difficult to find affordable housing that accepts companion animals [52,53]. Addressing issues like the shortage of affordable pet-friendly housing may be helpful to prevent situations in which older adults are faced with difficult decisions that may compromise their own well-being or that of their pet (e.g., having to choose between options that oppose their own well-being to that of their CAs). In Violet’s case, these were not manly because Jack was permitted in her apartment building. Nonetheless, future relocation to a dwelling that may not accept her dog was raised as a potential concern, by both Violet and her healthcare provider.

Lastly, another important finding is that the pet–owner relationship developed through pet ownership led to enhancing the older participant’s engagement in activities of daily living. Being responsible for the well-being of a living being motivates Violet to take care of herself, to remain healthy, and to be able to take care of her dog as long as possible. Sharing her daily life with Jack encourages her to carry out other daily activities independently, such as going out for shopping or taking Jack for walks. Similar findings have been reported by Johansson and colleagues (2014) in their qualitative study that explored community-dwelling older adults’ experiences with their CAs after a stroke, as their pets motivated them to handle daily activities due to a feeling of responsibility. Responsible pet ownership also involved helping their pet’s daily routine, which contributed to adding meaning to the older adults’ lives [30]. Notably, in Violet’s case, she has loved and cared for animals since her childhood. It is possible that being Jack’s caretaker and the human–animal bond formed with her dog enables her to pursue a significant lifelong role. For her, the reciprocal relationship with her dog, as well as providing adequate care to him is an important part of her role as a pet owner. Maintaining roles may provide a sense of continuity by linking the past to the future, which may facilitate the adaptation to transitions and buffer hardships [48]. The continuity theory, indeed, states that pursuing habits and a lifestyle may be one of the strategies that help people to adapt to the ageing process [49].

On one hand, pet ownership may provide older adults with an opportunity to pursue a meaningful role such as being a caregiver and may even be a way to sustain their independence in later years. Direct and indirect activities associated with pet care may nudge older adult owners into engaging in daily activities or leisure activities, as responsibilities may support physical, emotional, and financial independence [23,46,54]. On the other hand, ensuring a companion animal’s well-being is also a crucial part of responsible pet ownership, and should be considered when examining the role of pets in the older adults’ lives. For example, although Jack’s well-being was not at risk, interventions such as assistance from volunteers (e.g., from organizations like ElderDog Canada) to ensure he gets regular walks all year long could further promote Jack’s well-being [55]. This was not explored with the participants of this study, as it was not the purpose. However, future research could examine acceptable strategies for older adults and their healthcare providers to optimize the well-being of older adults and their companion animals, in cases where the challenges of pet ownership outweigh the benefits. Collaborations between community health services, animal welfare organizations, and animal health professionals may be a way to encourage such strategies. Research should help create policies that support human–animal relationships through pet ownership in the context of aging-in-place, while ensuring the well-being of both older adult pet owners and their companion animals.

### Limitations and Future Directions

This study has some limitations. First, although a total of four interviews were conducted with the participants, having more cases with older adults presenting different profiles would have increased their transferability to other community-dwelling older adults. Efforts were made to include more participants in the case, such as the older adult’s health aide, but this was not possible due to transfers of healthcare providers between health organizations, which were related to the pandemic. Second, the COVID-19 lockdown prevented direct observations of the participant’s home environment and interactions with her animal, which would have enriched the case. However, alternative ways were used to document this information and particular attention was taken to increase the credibility of the results (i.e., independent analysis, member checking). Although information related to the companion animal’s well-being was documented via the participants’ perspectives, the perspectives of an animal professional could have enriched the case. However, in the present study, the companion animal did not have an appointed health professional. Lastly, the single-case study approach does not establish causality. This should be noted, even though the aim of the study was to better understand and explore the benefits and challenges of pet ownership for older adults, rather than to find causality.

Future research should replicate this study with more cases to deepen the understanding of the role of pet ownership in the lives of community-dwelling older adults with disabilities. Such research is crucial to supporting human–animal relationships through pet ownership, for example by exploring how to strike a balance between the benefits and challenges of pet ownership (i.e., the demands of pet care vs. the older adult owner’s abilities). Interviews with healthcare providers, animal health professionals (e.g., veterinarians), animal behaviourists, and other community actors who are called to work with community-dwelling older adults may offer nuanced, contextually sensitive research.

## 5. Conclusions

Findings of this case study suggest that the benefits and challenges associated with pet ownership may be influenced by factors related to the characteristics of these individuals, their CAs, their environments, and their daily living activities. The perspectives of both older adults and their healthcare providers are important to gain a comprehensive understanding of the circumstances surrounding pet care. Being able to maintain human–animal relationships and to carry out meaningful roles, such as being a caregiver, may be an important part of healthy aging. Ultimately, future research should help develop strategies and/or policies that will aim to support pet ownership in the context of aging-in-place, while ensuring the well-being of both older adults and their companion animals.

## Figures and Tables

**Table 1 animals-11-02628-t001:** Example of interview questions.

Questions for the Older Adult
**Related to perceived benefits and challenges of taking care of Jack:** - Please describe any positive (or negative) aspects related to taking care of Jack and your emotions, if any. This was repeated for the different aspects of well-being, social life, daily routine, and health. - Please describe any positive (or negative) impacts for Jack, in his current living situation. **Related to the role of pet ownership in daily life:** - How does taking care of Jack positively (or negatively) influence your activities of daily living? Follow-up question: what do you mean by [participant’s answer]? - How does taking care of Jack influence the environments/spaces that you visit? - How would your daily routine be different if you did not take care of Jack daily? - Follow-up question: which activity would you engage or not engage in if Jack was not present?
**Questions for the Healthcare Provider**
**Related to perceived benefits and challenges of taking care of Jack:** - Please describe any positive (or negative) aspects for your client related to taking care of Jack and their emotions, if any. This was repeated for the different aspects of well-being, social life, daily routine, and health. Follow-up question: in what way is that aspect positive for Violet? - What do you perceive could be a potential risk for Violet related to taking care of Jack, if any? Follow-up question: how would you assess that risk (i.e., low, medium, high)? **Related to the role of pet ownership in daily life:** - What elements help Violet take care of Jack, if any? Follow-up question: how does this element help her take care of him?

## Data Availability

No new data were created or analyzed in this study. Data sharing is not applicable to this article.

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
