# Peer review of "Understanding the Benefits, Challenges, and the Role of Pet Ownership in the Daily Lives of Community-Dwelling Older Adults: A Case Study"

_animals, 2021, doi:10.3390/ani11092628_

Round 1
Reviewer 1 Report
This study is an interesting, well-conceived, and needed contribution to the field considering the scant research addressing the bond between humans with disabilities and their companion animals.
I have a few suggestions for details to add to clarify study protocol, expand on the themes, and assist future syntheses. I understand you may not have collected enough data to address some of these comments (e.g., Jack and his service/support animal status), and that is OK – the most important revision from my perspective are the comments tied to lines 136-150 and Table 1, as these are important for replication. All other suggestions are less important to address for publication, but would be nice to aid less-familiar members of the readership and future reviews and syntheses.
Lines 16-20 (Simple Summary): Be sure to clarify that this study was a case study using qualitative methods, as done in the abstract.
Line 37 (Abstract): Consider rephrasing “expose her to a low fall risk” to “introduce a slight fall risk” just in the abstract; I know the difference is subtle and wordier, but without reading the entire paper (as people are apt to do), I first took it to mean the dog reduced her existing fall risk.
Lines 34-56 (Introduction): Consider hedging some of these claims about the reported outcomes associated with pet ownership, as there is considerable variation in the methodological rigor of the cited studies from 16-24 and 25-29: not all these studies have causal designs, but “benefits” may read like “effects” to some readers as written.
Line 94 (Introduction): Clarify the definition of “community-dwelling” before this sentence to help readers like me who are less familiar with research on aging populations (i.e., does this mean the person is living in their own home rather than an Assisted Living facility or in long-term care?).
Line 120 (Methods – Participants): Clarify whether Jack was Violet’s companion animal, service dog, or emotional support animal if you can, to help aid future syntheses – it could be helpful for readers to know whether Jack has relevant service/assistance skills and whether she receives financial support and/or services tied to her disability benefits to help with Jack’s care.
Line 136 (Methods – Data Collection): Clarify whether Table 1 lists all the interview questions or a smaller subset of the total for each informant. Were follow-up questions asked of each informant to expound on initial responses? How many separate interviews were conducted and how long was each interview?
Lines 142-144 (Methods – Data Collection): Clarify a) whether the questions about the home environment, CA interactions, and socio-demographic data were collected as a part of the described interviews or via survey, and b) to what extent the information collected on each of these variables is represented in the Case Summary and Themes presented below.
Line 150 (Table 1) Consider editing Table 1 to indicate which questions were asked of the healthcare provider, which questions were asked of Violet, and which questions were repeated for both informants. Does the inclusion of “(emotions, well-being, social life, daily routine, etc.)” mean that this question was repeated for each of these aspects? If so, what other variables were captured in the “etc.”?
Line 152 (Methods – Data Analysis) This section is well-written with details clearly explained throughout.
Line 172 (Results): The results are interesting and relevant to the research questions. However, the mention of the healthcare provider's input is inconsistent until the later themes (e.g., Physical benefits and challenges); clarify for each theme (e.g., Jack’s influence on Violet’s mood) whether and in what way the healthcare provider’s responses corroborated with or contradicted Violet’s perspective.
Line 173 (Results – Case Summary): This is a thorough and interesting description of Violet, her circumstances, and the way Jack came into her life. The only thing I could think to add to this section is a few more details about Jack's characteristics (if available), such as his size (weight, etc.) and energy requirements, since it is mentioned later that Jack can introduce a fall risk when he is underfoot or picked up, and that their daily walks take a pause in the winter (as a reader I wondered “does this influence his behavior in the winter? Do they compensate with indoor play during that time?”).
Line 217 (Results – Psychological Benefits and Challenges): This section was enjoyable to read, and the quote powerfully exemplifies Jack's influence on Violet's mood. What I feel is missing in this section are quotes/details that highlight her feelings of safety - does she mean emotionally safe or physically safe? What aspect of Jack's presence or behavior contribute to her feeling safe?
Line 236 (Results – Psychological Benefits and Challenges): This section and quote are heart-warming and represent the theme well.
Line 320 (Results – Well-being of the Companion Animal): Clarify whether information was reported about managing Jack's waste, as the comment on taking him outside (weather permitting) raised these questions for me: Does Jack use puppy pads when Violet can’t take him outside? Can Violet manage that waste on her own, or does she have others help pick it up (both indoors and outside)? In addition, do Jack and Violet engage in indoor activities when he cannot be walked?
Line 330 (Results – Meaningful Activity): While I hear Violet's perspective that she does not engage with Jack to stave off social isolation or loneliness and the healthcare provider’s claim that Violet is not socially isolated, as a reader I sensed a disconnect between these claims and Violet’s reported need to "keep busy" or have "another presence" as mentioned in the quote, and how lonely she reported she would be without Jack as explained in other sections. I would recommend exploring this disconnect/tension either in the findings or the discussion if you have insights on this.
Line 382 (Results – Providing Purpose and Routine): Consider mentioning that "now" refers to "during the COVID-19 pandemic," if that is the case, to clarify for future, post-pandemic readers.
Line 453 (Discussion) Consider also discussing Jack's characteristics (e.g., size, temperament, energy level) and their role in the fit of the Jack-Violet relationship, as this dynamic may have played out differently with a large, high-energy dog (e.g., German Shorthaired Pointer) or a dog that requires more maintenance.
Line 524 (Discussion – Limitations and Future Directions) These are good points about the additional data that could have been collected in this study. Consider also mentioning the limitation of the case study approach in establishing benefits/effects, as this single-case design does not establish causality between pet ownership and the benefits perceived by Violet and her healthcare provider in this circumstance, and may not extrapolate to other community-dwelling older adults depending on the human, animal, and contextual factors you have outlined in other sections.
Author Response
Point 1: This study is an interesting, well-conceived, and needed contribution to the field considering the scant research addressing the bond between humans with disabilities and their companion animals.
I have a few suggestions for details to add to clarify study protocol, expand on the themes, and assist future syntheses. I understand you may not have collected enough data to address some of these comments (e.g., Jack and his service/support animal status), and that is OK – the most important revision from my perspective are the comments tied to lines 136-150 and Table 1, as these are important for replication. All other suggestions are less important to address for publication, but would be nice to aid less-familiar members of the readership and future reviews and syntheses.
Response 1: Thank you for the positive feedback. We have modified lines 136-150 and Table 1 as recommended. We have also addressed other suggestions, when possible (see the responses below).
Point 2: Lines 16-20 (Simple Summary): Be sure to clarify that this study was a case study using qualitative methods, as done in the abstract.
Response 2: We have clarified this by adding: “This qualitative case study explores […]”
Point 3: Line 37 (Abstract): Consider rephrasing “expose her to a low fall risk” to “introduce a slight fall risk” just in the abstract; I know the difference is subtle and wordier, but without reading the entire paper (as people are apt to do), I first took it to mean the dog reduced her existing fall risk.
Response 3: We agree that it could be misinterpreted and have rephrased it as suggested. Thank you.
Point 4: Lines 34-56 (Introduction): Consider hedging some of these claims about the reported outcomes associated with pet ownership, as there is considerable variation in the methodological rigor of the cited studies from 16-24 and 25-29: not all these studies have causal designs, but “benefits” may read like “effects” to some readers as written.
Response 4: We have nuanced some claims (line 50-52):
- “Human-animal interactions (HAIs) may be an avenue worth exploring to support the health and well-being of this population, as research about the positive psychological, social, and physical impacts of HAIs is promising [8–14].”;
- Relating to psychological and social benefits, recent studies suggest that pet ownership may improve well-being, life satisfaction and happiness, as well as decrease loneliness and social isolation, depressive symptoms and anxiety.”
- “It may also increase levels of physical activity and/or walking of older adults pet owners.”
Point 5: Line 94 (Introduction): Clarify the definition of “community-dwelling” before this sentence to help readers like me who are less familiar with research on aging populations (i.e., does this mean the person is living in their own home rather than an Assisted Living facility or in long-term care?).
Response 5: We clarified the definition (line 94-97): “For this study, community-dwelling older adults included older adults living in their homes or in an assisted living facility, but excluded those living in nursing homes (i.e., needing continuous medical care).”
Point 6: Line 120 (Methods – Participants): Clarify whether Jack was Violet’s companion animal, service dog, or emotional support animal if you can, to help aid future syntheses – it could be helpful for readers to know whether Jack has relevant service/assistance skills and whether she receives financial support and/or services tied to her disability benefits to help with Jack’s care.
Response 6: We have added this sentence to clarify (line 126-129): “Jack is Violet’s companion animal and has not received any training to acquire specific skills (i.e., he is not considered a service dog or emotional support animal). Violet does not receive any financial support or services related to her disabilities to help with Jack’s care.”
Point 7: Line 136 (Methods – Data Collection): Clarify whether Table 1 lists all the interview questions or a smaller subset of the total for each informant. Were follow-up questions asked of each informant to expound on initial responses? How many separate interviews were conducted and how long was each interview?
Response 7:
- In lines 144-145 it was already mentioned “Table 1 provides examples of interview questions for both participants.” It is custom to ask follow-up questions when conducting interviews. We added examples of follow-up questions to Table 1 and mentioned that “some interview questions and follow-up questions”.
- We specified how many interviews were conducted and the duration (Line 148-150): “Two interviews were carried out in April 2020 via visio-conference with the healthcare provider and by telephone with the older adult, due to the COVID-19 context. Each interview lasted from 60 to 90 minutes.”
Point 8: Lines 142-144 (Methods – Data Collection): Clarify a) whether the questions about the home environment, CA interactions, and socio-demographic data were collected as a part of the described interviews or via survey, and b) to what extent the information collected on each of these variables is represented in the Case Summary and Themes presented below.
Response 8: We clarified this in lines 154-159: “Sociodemographic data about age, gender, education, years of clinical experience for the community healthcare provider and pet characteristics (age, species, number of years liv-ing with the older adult, pet care activities, health condition) were also collected during the interviews to provide contextual information. This information is reported in the results section to the extent that the confidentiality of the participant was kept.”
Point 9: Line 150 (Table 1) Consider editing Table 1 to indicate which questions were asked of the healthcare provider, which questions were asked of Violet, and which questions were repeated for both informants. Does the inclusion of “(emotions, well-being, social life, daily routine, etc.)” mean that this question was repeated for each of these aspects? If so, what other variables were captured in the “etc.”?
Response 9: We have edited the table to align with the suggestions (see Table 1, line 160).
Point 10: Line 152 (Methods – Data Analysis) This section is well-written with details clearly explained throughout.
Response 10: Thank you for the positive feedback.
Point 11: Line 172 (Results): The results are interesting and relevant to the research questions. However, the mention of the healthcare provider's input is inconsistent until the later themes (e.g., Physical benefits and challenges); clarify for each theme (e.g., Jack’s influence on Violet’s mood) whether and in what way the healthcare provider’s responses corroborated with or contradicted Violet’s perspective.
Response 11: We clarified this in:
- lines 234-236: “The main finding regarding the psychological benefits is that Jack’s presence in Violet’s daily life has a positive influence on her emotions—an aspect that both Violet and her healthcare provider agree on.”;
- lines 245-246: “According to both Violet and her healthcare provider, Jack also helps Violet channel stress and anxiety, which helps calm her down and weather difficult emotions.”;
- line 272: “This aspect was only mentioned by Violet.”;
- line 282-283: “Both participants agree that being obliged to part with Jack under such circumstances would be difficult and would bring up negative emotions.”
Point 12: Line 173 (Results – Case Summary): This is a thorough and interesting description of Violet, her circumstances, and the way Jack came into her life. The only thing I could think to add to this section is a few more details about Jack's characteristics (if available), such as his size (weight, etc.) and energy requirements, since it is mentioned later that Jack can introduce a fall risk when he is underfoot or picked up, and that their daily walks take a pause in the winter (as a reader I wondered “does this influence his behavior in the winter? Do they compensate with indoor play during that time?”).
Response 12: We added Jack’s size (line190): “Jack is a small adult dog, weighing approximately 12 lbs and he is sturdy, according to Violet. She reports that these characteristics are an adequate fit for her and enable her to maintain her balance while picking him up.” We have not added the other information about energy requirements since we did not collect it.
Point 13: Line 217 (Results – Psychological Benefits and Challenges): This section was enjoyable to read, and the quote powerfully exemplifies Jack's influence on Violet's mood. What I feel is missing in this section are quotes/details that highlight her feelings of safety - does she mean emotionally safe or physically safe? What aspect of Jack's presence or behavior contribute to her feeling safe?
Response 13: The interviewer asked the participant in which way her dog made her feel safe, but the older adult was not able to put words on how he made her feel safe. We suspect that he makes her feel “emotionally safe”, but prefer not labelling it to avoid misinterpreting the data. We thus added this quote to illustrate in the participant’s own words how he makes her feel safe (236-238): “I feel safe, in the house. That’s very important for me because I don’t like the dark-ness, I don’t like when there is wind, when it thunders…so Jack makes me feel safe, you know? […] At least I’m not alone in the house, there is a presence.”
Point 14: Line 236 (Results – Psychological Benefits and Challenges): This section and quote are heart-warming and represent the theme well.
Response 14: Thank you for the positive feedback.
Point 15: Line 320 (Results – Well-being of the Companion Animal): Clarify whether information was reported about managing Jack's waste, as the comment on taking him outside (weather permitting) raised these questions for me: Does Jack use puppy pads when Violet can’t take him outside? Can Violet manage that waste on her own, or does she have others help pick it up (both indoors and outside)? In addition, do Jack and Violet engage in indoor activities when he cannot be walked?
Response 15: We have clarified this by adding a sentence (line 304-305): “Also, Jack uses puppy pads inside the home, which enables Violet to take care of her dog’s needs independently even when she cannot take him outside.”
Point 16: Line 330 (Results – Meaningful Activity): While I hear Violet's perspective that she does not engage with Jack to stave off social isolation or loneliness and the healthcare provider’s claim that Violet is not socially isolated, as a reader I sensed a disconnect between these claims and Violet’s reported need to "keep busy" or have "another presence" as mentioned in the quote, and how lonely she reported she would be without Jack as explained in other sections. I would recommend exploring this disconnect/tension either in the findings or the discussion if you have insights on this.
Response 16: This is indeed an interesting aspect and merits further research. We have not explored this aspect in detail with the participants, as it was not the focus of the study, but we have added a sentence to note this tension (line 469-475): “It is interesting to note that Violet’s healthcare provider does not consider Violet as being socially isolated and does not perceive pet ownership as a way to compensate a lack of social relationships, in Violet’s case. Such findings may indicate the importance of the human-animal bond that is formed through pet ownership despite an older adult being socially connected. Human-animal bonds may fulfill different needs in daily life, even when an older adult maintains social relationships, which is an aspect that merits further research.”
Point 17: Line 382 (Results – Providing Purpose and Routine): Consider mentioning that "now" refers to "during the COVID-19 pandemic," if that is the case, to clarify for future, post-pandemic readers.
Response 17: We have added the information (line 397-398): “…, like they do now [during the COVID-19 pandemic].”
Point 18: Line 453 (Discussion) Consider also discussing Jack's characteristics (e.g., size, temperament, energy level) and their role in the fit of the Jack-Violet relationship, as this dynamic may have played out differently with a large, high-energy dog (e.g., German Shorthaired Pointer) or a dog that requires more maintenance.
Response 18: This was mentioned in the sentence (“Violet and Jack’s case may be an example of how an adequate fit between the characteristics of the older adult, their CA and their environment may maximize the benefits related to pet care”), but we have added a sentence to provide further detail (lines 476-478): “According to Violet, her dog’s physical characteristics (i.e., small size, sturdy stature) fit well with her own characteristics. This may also play a role in managing the fall risk.”
Point 19: Line 524 (Discussion – Limitations and Future Directions) These are good points about the additional data that could have been collected in this study. Consider also mentioning the limitation of the case study approach in establishing benefits/effects, as this single-case design does not establish causality between pet ownership and the benefits perceived by Violet and her healthcare provider in this circumstance, and may not extrapolate to other community-dwelling older adults depending on the human, animal, and contextual factors you have outlined in other sections.
Response 19: Although the aim of the study was not to establish causality, we understand that with our study aim it could be argued that we are looking for a “link” between pet ownership and benefits. Therefore, we added this sentence to add the limitation (lines 565-567): “Lastly, the single-case study approach does not establish causality. This should be noted, even though the aim of the study was to better understand and explore the benefits and challenges of pet ownership for older adults, rather than to find causality.”
Reviewer 2 Report
The case study addresses a relevant subject, indicated to deepen the understanding of human-non-human animal interaction from the perspective of its older tutor inserted in the community and from the perspective of the healthcare manager.
The methodology used is adequate and described in a clear and rigorous way. The results obtained are relevant and enable the exploration and deepening of knowledge in this field.
Author Response
Point: The case study addresses a relevant subject, indicated to deepen the understanding of human-non-human animal interaction from the perspective of its older tutor inserted in the community and from the perspective of the healthcare manager.
The methodology used is adequate and described in a clear and rigorous way. The results obtained are relevant and enable the exploration and deepening of knowledge in this field.
Response: Thank you very much for your positive feedback, we appreciate it.
Reviewer 3 Report
Overall, I found this paper to be a very interesting and thought-provoking read and congratulate the authors on this good work. I think it has the potential to inspire future research in this important and timely area of study. The most significant suggestion I have for the authors is to consider not labeling the subject as ‘frail’ throughout the paper. If the authors want to keep the ‘frail’ label in the title and throughout the paper, I would suggest including a paragraph describing frailty syndrome and its consequences in the introduction to make this more of a focus of the paper. I think describing the participant as a community-dwelling older adult with physical disabilities would actually be a better general descriptor, and simply making it clear that she has frailty in the Case Summary section (3.1). Below I will detail a few other suggestions for the authors to consider that I believe would strengthen the manuscript.
Abstract: Clarify the aim statement. Specifically the sentence beginning on line 32 that reads, ‘This case study aims to better understand the role of pet ownership [in what] and explore the benefits and challenges of owning a pet [in older adulthood?]’
Introduction:
-I believe the summary of research on the psychological and social benefits of pet ownership (lines 54-57) is overstated. The research is mostly cross-sectional and findings have been inconclusive. I would encourage the authors to soften their tone- pets may help in many ways and research in this area is promising, but it is far from conclusive.
Table 1:
-Perhaps consider separating this Table into 2 sections- questions for the older adult and questions for the health care provider- with sub-headings within the sections (i.e., benefits and challenges of taking care of dog, role of pet ownership in daily life)
Table 2:
-I would suggest deleting Table 2. I do not think it adds significantly to the paper.
Results:
-Did the participant live in a rural, suburban or urban area? This additional context would be helpful, if available.
-Please provide more information on the PRISMA-7 measure and what a score of 4 indicates. This is especially important given the authors emphasis on the participant’s frailty status.
-The authors should add details re: the statement on line 188 ‘… she showed no signs of cognitive impairment’. How was this determined? This sounds very discretionary. Is cognition evaluated in the PRISMA-7? Or did the authors formally assess cognitive impairment in anyway? Perhaps at minimum add that she has no diagnosis of mild cognitive impairment or dementia.
-I would encourage the authors to remove section 3.3. It does not seem valid to include an assessment of the dog’s health and well-being based on the owner’s opinion and the owner’s healthcare provider’s opinion- a veterinarian would be needed to provide this insight, as the authors note in their Limitations section.
Finally, I also think the writing needs to be improved in general- there were grammatical errors throughout the paper.
Author Response
We have modified the manuscript according to the suggestions. Thank you for the constructive feedback.
Point 1: Overall, I found this paper to be a very interesting and thought-provoking read and congratulate the authors on this good work. I think it has the potential to inspire future research in this important and timely area of study. The most significant suggestion I have for the authors is to consider not labeling the subject as ‘frail’ throughout the paper. If the authors want to keep the ‘frail’ label in the title and throughout the paper, I would suggest including a paragraph describing frailty syndrome and its consequences in the introduction to make this more of a focus of the paper. I think describing the participant as a community-dwelling older adult with physical disabilities would actually be a better general descriptor, and simply making it clear that she has frailty in the Case Summary section (3.1). Below I will detail a few other suggestions for the authors to consider that I believe would strengthen the manuscript.
Response 1: We agree with this suggestion and we have removed the label “frail” throughout the paper and described the participant as having physical disabilities, since the frailty status was not the focus of the article.
Point 2: Abstract: Clarify the aim statement. Specifically the sentence beginning on line 32 that reads, ‘This case study aims to better understand the role of pet ownership [in what] and explore the benefits and challenges of owning a pet [in older adulthood?]’
Response 2: Line 32-34; We have clarified the statement “This case study aims to better understand the role of pet ownership in the daily lives of older adults and explore the benefits and the challenges of owning a pet in older adulthood.
Point 3: Introduction:
-I believe the summary of research on the psychological and social benefits of pet ownership (lines 54-57) is overstated. The research is mostly cross-sectional and findings have been inconclusive. I would encourage the authors to soften their tone- pets may help in many ways and research in this area is promising, but it is far from conclusive.
Response 3: We have modified the sentence to soften the tone (line 50-52): “Human-animal interactions (HAIs) may be an avenue worth exploring to support the health and well-being of this population, as research about the positive psychological, social, and physical impacts of HAIs is promising [8–14]”
Point 4: Table 1:
-Perhaps consider separating this Table into 2 sections- questions for the older adult and questions for the health care provider- with sub-headings within the sections (i.e., benefits and challenges of taking care of dog, role of pet ownership in daily life)
Response 4: We have edited the table to align with the suggestions (see Table 1, line 160).
Point 5: Table 2:
-I would suggest deleting Table 2. I do not think it adds significantly to the paper.
Response 5: We find that this table illustrates well how Violet’s and her dog’s, Jack, daily routines are integrated. We removed it from the paper, but propose to keep it as an appendix, if possible. We can send a file with Table 2 if this option is acceptable.
Point 6: Results:
-Did the participant live in a rural, suburban or urban area? This additional context would be helpful, if available.
Response 6: We added this information (line 176-177): “Violet and her dog Jack live in an urban area, in a one-bedroom rent-subsidized apartment.”
Point 7: -Please provide more information on the PRISMA-7 measure and what a score of 4 indicates. This is especially important given the authors emphasis on the participant’s frailty status.
Response 7: We removed the reference to the frailty status and the score, which also align with point 1 (line 190-200): “According to the PRISMA-7, a screening tool that identifies older adults at risk for moderate to severe disabilities, she has significant disabilities [43]”
Point 8: -The authors should add details re: the statement on line 188 ‘… she showed no signs of cognitive impairment’. How was this determined? This sounds very discretionary. Is cognition evaluated in the PRISMA-7? Or did the authors formally assess cognitive impairment in anyway? Perhaps at minimum add that she has no diagnosis of mild cognitive impairment or dementia.
Response 8: We added details to respond to the suggestion (200-202): “According to her healthcare provider, Violet has adequate cognitive functioning, which was also the case during the interviews (i.e., she showed no signs of cognitive impairment according to the clinical judgement of the interviewer).”
Point 9: -I would encourage the authors to remove section 3.3. It does not seem valid to include an assessment of the dog’s health and well-being based on the owner’s opinion and the owner’s healthcare provider’s opinion- a veterinarian would be needed to provide this insight, as the authors note in their Limitations section.
Response 9: We agree that an assessment of the dog’s health by a veterinarian is ideal to provide an objective insight of the companion animal’s health. However, the older adult participant did not have a regular veterinarian for her dog. Although the owner’s and the owner’s healthcare provider’s opinions are subjective, we feel that it is important to document them. Since the owner is the caretaker of the animal and provides daily care, their perspective is important to understand how they understand the care of an animal. Furthermore, our research also considers the well-being of the animal from different perspectives. We have thus kept this section and to reflect that this is a subjective assessment, we have added: “From both Violet and her healthcare provider’s perspectives, Jack received adequate care from Violet.” We also added that these statements were according to the participants.
Point 10: Finally, I also think the writing needs to be improved in general- there were grammatical errors throughout the paper.
Response 10: The manuscript was reviewed by a person a proficient in English and modified accordingly.